# Chemical Profiling, Formulation Development, *In Vitro* Evaluation and Molecular Docking of *Piper nigrum* Seeds Extract Loaded Emulgel for Anti-Aging

**DOI:** 10.3390/molecules27185990

**Published:** 2022-09-14

**Authors:** Muhammad Yousuf, Haji Muhammad Shoaib Khan, Fatima Rasool, Kashif ur Rehman Khan, Faisal Usman, Bilal Ahmad Ghalloo, Muhammad Umair, Ahmad O. Babalghith, Muhammad Kamran, Rana Muhammad Aadil, Soad K. Al Jaouni, Samy Selim, Sameh A. Korma, Carlos Adam Conte-Junior

**Affiliations:** 1Department of Pharmaceutics, Faculty of Pharmacy, The Islamia University of Bahawalpur, Bahawalpur 63100, Pakistan; 2Department of Pharmacy, University College of Pharmacy, University of the Punjab, Lahore 54590, Pakistan; 3Department of Pharmaceutical Chemistry, Faculty of Pharmacy, The Islamia University of Bahawalpur, Bahawalpur 63100, Pakistan; 4Department of Pharmaceutics, Faculty of Pharmacy, Bahaud-Din-Zakariya University, Multan 60700, Pakistan; 5Department of Food Science and Engineering, College of Chemistry and Engineering, Shenzhen University, Shenzhen 518060, China; 6College of Food Science and Technology, Nanjing Agriculture University, Nanjing 210095, China; 7Medical Genetics Department, College of Medicine, Umm Al-Qura University, Makkah 24382, Saudi Arabia; 8Department of Parasitology and Microbiology, Faculty of Veterinary and Animal Sciences, Pir Mehr Ali Shah Arid Agriculture University, Rawalpindi 46000, Pakistan; 9National Institute of Food Science and Technology, University of Agriculture, Faisalabad 38000, Pakistan; 10Department of Hematology/Oncology, Yousef Abdulatif Jameel Scientific Chair of Prophetic Medicine Application, Faculty of Medicine, King Abdulaziz University, Jeddah 21589, Saudi Arabia; 11Department of Clinical Laboratory Sciences, College of Applied Medical Sciences, Jouf University, Sakaka 72388, Saudi Arabia; 12Department of Food Science, Faculty of Agriculture, Zagazig University, Zagazig 44519, Egypt; 13Center for Food Analysis (NAL), Technological Development Support Laboratory (LADETEC), Federal University of Rio de Janeiro (UFRJ), Cidade Universitária, Rio de Janeiro 21941-598, Brazil

**Keywords:** *Piper nigrum* extract, emulgel, antioxidant, anti-aging, bioactive compounds, terpenoids, polyphenol, anti-tyrosinase, formulation development

## Abstract

Emulgel is a new innovatory technique for drug development permitting controlled release of active ingredients for topical administration. We report a stable emulgel of 4% *Piper nigrum* extract (PNE) prepared using 80% ethanol. The PNE-loaded formulation had an antioxidant activity of 84% and tyrosinase inhibition was 82%. Prepared formulation rendered spherical-shaped globules with high zeta potential (−45.5 mV) indicative of a stable system. Total phenolic contents were 58.01 mg GAE/g of dry extract whereas total flavonoid content was 52.63 mg QE/g of dry extract. Sun protection factor for PNE-loaded emulgel was 7.512 and formulation was stable without any evidence of physical and chemical changes following 90 days of storage. Gas chromatography-mass spectroscopy (GC-MS) revealed seventeen bioactive compounds in the PNE including monoterpenoids, triterpenoids, a tertiary alcohol, fatty acid esters, and phytosterols. In silico studies of GC-MS identified compounds show higher binding affinity in comparison to standard kojic acid indicating tyrosinase inhibition. It can be concluded that PNE-loaded emulgel had prominent antioxidant and tyrosinase inhibition and can be utilized as a promising topical system for anti-aging skin formulation.

## 1. Introduction

Since the start of human life on Earth, people have turned to medicinal plants to treat a wide range of illnesses. The quest to synthesize medicinally relevant chemicals and their derivatives in laboratories began in the 20th century as human civilization advanced and the isolation of medicinally important molecules increased. Despite the fact that many active medicinal substances have been created synthetically, the value of plants cannot be understated, which is why they are continually searched for new molecules with biological significance [1]. There is an increasing trend in the use of natural ingredients for skin care purposes, and they are becoming more prevalent in modern formulations. In contrast to synthetic products, natural compounds of herbal origin are well documented to demonstrate a variety of biological functions [2]. Emulgel-containing natural extracts have gained more importance owing to less toxicity, greater therapeutic value, and cosmetic effects [3]. Moreover, people’s trends towards natural ingredients in cosmetics increase the significance of cosmetic products with natural extracts [4]. Emulsions are bi-phasic topical drug delivery systems and are frequently used for the delivery of potent and effective phytochemical ingredients to the skin in the treatment of various skin diseases, intended to provide beneficial protective effects for degenerative skin. The arrangement of two immiscible liquids in which one liquid is dispersed (dispersed phase) into a second liquid (continuous phase) in the presence of an emulsifying agent is called an emulsion [5]. Mainly, there are two types of emulsion systems, i.e., w/o (water is dispersed phase and oil is continuous phase) and o/w (oil is dispersed phase and water is continuous phase). The gel is a system of two phases in which liquid molecules are dispersed in a second solid medium [6]. Emulgels are widely used in pharmaceutical industries in topical drug delivery. In some conditions, for little enhancement of the bioavailability and masking of unpleasant taste and odor, the emulgel system is a better choice [5]. Emulgel for skin usage has some favorable qualities such as being thixotropic, greaseless, easily spreadable, easily removable, emollient, non-staining, water-soluble, longer shelf life, bio-friendly, clear and pleasing appearance [6]. Cosmetology has gained more importance with products containing natural ingredients and less toxic effects in comparison to synthetic ingredients containing products with more toxic effects [7].

Spices are dietary ingredients used every day to increase the flavor and perception of the human diet. Among spices, *Piper nigrum* is widely used across the world, and so it is titled as the “king of spices” [5]. Additionally, in Turkish folk medicines, *P. nigrum* seeds were known as hunger drugs. It helps to improve digestion, increase hunger, and helps in therapy for sneezing, coughing, dyspnea, disease of tonsils, piles, recurrent temperature, malnutrition, larvae, rheumatism, and wound healing. It also possesses significant antimicrobial, pain-relieving, and anti-inflammatory activity. *P. nigrum* belongs to the family Piperaceae. *P. nigrum* constituents include piperine, phenols, alkaloids, amides, proteins, safrol, terpenes, and dietary fibers [6]. Owing to phenolic content in *P. nigrum,* it shows antioxidant activity and can be an easily reachable basis of natural radical scavenging activity in the nutrition and cosmetic industry. In addition to these uses, *P. nigrum* is used in perfumery as a flavor, preservative, and medicinal moiety. Due to the related physiological activities, this species has significant commercial, economic and therapeutic potential [7]. The different free radical scavenging mechanisms of *P. nigrum* extracts may be accredited to their firm hydrogen providing capacity, metal chelating capacity, and usefulness as a decent hunter of hydrogen peroxide, superoxide, and free radicals [8]. *P. nigrum* and piperine have a wide range of pharmacological effects, including antitumor, antidepressant, anti-asthmatic, anti-hypertensive, and antiplatelet properties, as well as anti-inflammatory, antimicrobial, antioxidant, hepatoprotective, anti-diarrheal, immunomodulatory, anticonvulsant, and analgesic properties [8].

The objective of this study was to formulate a stable emulgel of *P. nigrum* seed extract, followed by *in vitro* and in silico characterization for anti-aging effects. The *in vitro* characterization was carried out by organoleptic evaluation (color, liquefaction, phase separation, microbial growth, and centrifugation), pH, and electrical conductivity, which were carried out for both control and extract-containing formulations at various storage parameters (8 ± 1 °C, 25 ± 1 °C, 40 ± 1 °C, and 40 ± 1 °C + 75 ± 1% relative humidity (RH)) for a period of three months. Skin tolerance test, spreadability, drug content determination, and *in vitro* release studies were performed. The in silico characterization was performed by molecular docking of gas chromatography-mass spectroscopy (GC-MS) identified compounds with tyrosinase. All the results of the performed analysis suggest that formulated stable emulgel could be a potential candidate for anti-aging preparations.

## 2. Results

### 2.1. Total Phenolic Contents

The results were expressed as milligrams of gallic acid equivalents per gram of extract (mg GAE/g dry extract). The standard curve of gallic acid was plotted, and total phenolic content (TPC) was calculated by regression constant. The TPC of *P. nigrum* extract was 58.01 mg GAE/g of dry extract.

### 2.2. Total Flavonoids Contents

Total flavonoid content (TFC) was expressed as milligrams of quercetin equivalents per gram of extract (mg QE/g of dry extract). In the present study, the TFC in plant extract was 52.63 mg QE/g of dry extract.

### 2.3. Antioxidant Potential

The antioxidant activity of *P. nigrum* abstract was 84% at a concentration of 1 mg/mL determined by DPPH (2,2-diphenyl-1-picrylhydrazyl) in ethanol against ascorbic acid as a standard, as depicted in Figure 1A.

### 2.4. Tyrosinase Enzyme Inhibition Activity

The tyrosinase inhibition activity for *P. nigrum* extracts was calculated to be 82% and is presented graphically in Figure 1B.

### 2.5. Gas Chromatography-Mass Spectroscopy (GC-MS)

Outcomes are described in Figure 2 and Table 1, which reveal the presence of major components in the *P. nigrum* extract. Table 1 presents the retention time, peak area (%), compound name, molecular formula, molecular weight, compound nature, similarity index, and pharmacological activity for each major component found in the *P. nigrum* extract. The biological activities of *P. nigrum* extract are shown according to Duke’s phytochemical and ethnobotanical databases. GC-MS analysis revealed, several phyto-components such as monoterpenoids, terpineol, sesquiterpenoid, a tertiary alcohol, long-chain fatty acids, fatty acid ester, acyclic diterpenoids, alkaloids, methylated phenol, and phytosterol compounds in the seed extract of *P. nigrum*. All phyto-components in *P. nigrum* seed extract were qualitatively analyzed using gas chromatography. GC-MS and results of the main compounds are listed in Table 1.

### 2.6. Stability Studies

#### 2.6.1. Organoleptic Properties

During the study of 90 days, no significant change was observed in color, odor, phase separation, and liquefaction in both formulations, i.e., base and formulation containing plant extract at any temperature and RH. Table 2 shows the organoleptic characteristics and microbial development in the formulation at different temperatures.

#### 2.6.2. pH

The pH of the control and test formulation were 6.1 and 5.9, respectively, on the first day and were 5.6 and 5.4, respectively, following three months of study. This change in the pH of the base and extract formulation is shown in Figure 3.

#### 2.6.3. Microscopy

Throughout the study period, emulsion globules were found to be spherical in both the placebo and the *P. nigrum* extract-containing formulations. Moreover, there was a minor increasing trend in globule size with the passage of time. There was no modification in the morphology of globules, and globules were persistently identical and spherical in shape during the entire duration of the study, as presented in Figure 4.

### 2.7. Electrical Conductivity

A rise in electrical conductivity in both the placebo and test formulations was recorded. This increase in electrical conductivity was more noticeable in both formulations at high temperatures and high RH (40 ± 1 °C and 40 ± 1 °C + 75 ± 1% RH), but it was not statistically significant (*p* > 0.05). The result is depicted in Figure 5 for the placebo and plant extract formulations.

### 2.8. Rheological Evaluation

Figure 6 and Figure 7 represent the rheograms of each formulation kept at various parameters of temperature and humidity, such as 8 ± 1 °C, 25 ± 1 °C, 40 ± 1 °C, and 40 ± 1 °C + 75 ± 1% RH.

### 2.9. Centrifugation Test

There was no phase separation for both formulations at any time during the entire study duration, indicating the stability of the formulations. Emulgel stability is basically due to the proper homogenization at preparation time, so the globule size should be as small as possible and fully dispersed in the gel medium. The results of the centrifugation test are presented in Table 2.

### 2.10. Sun Protection Factor (SPF)

In the present research, the formulation containing *P. nigrum* extract exhibited an SPF value of 7.5116.

### 2.11. Zeta Potential and Globule Size

The result of particle size and zeta potential is shown in Figure 8 and Table 3. According to the zeta potential distribution graph, the zeta potential of *P. nigrum* extract containing emulgel formulation is −45.5 mV whereas the size is 675.9 nm, which is shown by the size distribution intensity graph. A higher value of zeta potential was indicative of a stable system with minimal chances of coalescence.

### 2.12. Skin Irritation (Tolerance) Test

The findings revealed that after 72 h of application, neither the *P. nigrum* emulgel nor the plain gel samples caused any noticeable morphological changes on the normal rat skin, such as erythema, papules, wrinkling, or dermatitis, indicating that neither had the potential to cause irritation, hypersensitivity, or toxicity reactions.

### 2.13. Spreadability

The spreadability test revealed that the control and the formulation had a spreadability that varied from 70 g.cm/min to 110 g.cm/min, respectively.

### 2.14. Drug Content Determination

The formulations’ drug content was calculated using a standard plot, and the result was 85.47%.

### 2.15. In Vitro Release

The results of the *in vitro* release of *P. nigrum* bioactive components from *P. nigrum* extract formulation (PNEF) and *P. nigrum* control formulation (PNCF) were 31.90% and 22.50%, respectively, as shown in Figure 9. The percent cumulative drug release of PNEF and PNCF was 31.90% and 22.50%, respectively, during the 12 h of the study. Both PNEF and PNCF showed non-linear biphasic drug release. Drug release from PNEF was initially rapid, about 18.10% during the first 4 h of the study, and then slower, up to 29.20% in the next 8 h. PNCF released about 10.20% of the drug in the first 4 h of the study and about 20.7% of the drug was released slowly in the next 8 h of study.

### 2.16. Molecular Docking for Tyrosinase Enzyme

Four compounds identified by GC-MS, i.e., γ-tocopherol, campesterol, phytol, myristic acid, and standard kojic acid, were selected to quantify the binding affinity and binding interaction with the target tyrosinase enzyme. As shown in Table 4, the binding affinity of γ-tocopherol and campesterol (6.6 and 6.5 Kcal/mol, respectively) was greater than the standard kojic acid binding affinity (5.4 Kcal/mol). This exhibited the vital part of *P. nigrum* in tyrosinase inhibition. Graphically, the binding affinities of compounds are shown in Figure 10 and Figure 11.

## 3. Discussion

Phenolic compounds are among those secondary metabolites that contain heterogeneous structures and are considered very important constituents of plants. They play a pivotal role in the prevention of plants against stressful conditions, including pollutants, extremes of temperature, ultraviolet radiation, and infections from pathogens. Moreover, it is reported from various studies that phenolic constituents are linked with free radical scavenging activities. These compounds also show a vital role in stabilizing fat peroxidation. In a recent study, total phenolic content in *P. nigrum* seed extract was calculated using a standard regression line for gallic acid (y = 0.0078x + 0.3366, R² = 0.9991). Previous studies conducted to assess the phenolic content of *P. nigrum* seed extracts showed that they are rich in phenolic content. According to current studies, an association between phenols and free radical scavenging activities has been established in numerous plants [15].

Herbs and spices (natural sources) are known to have phyto-antioxidant compounds such as polyphenols, flavonoids, and phenolic components that have the capability to avoid the damaging properties of free radicals owing to their antioxidant effects. Numerous techniques can be used for the assessment of the unpaired radical hunting capability of antioxidants. The electron-giving capability of natural plant abstracts was determined using the DPPH method [12]. The results of the current study indicated that *P. nigrum* seed extracts possess excellent antioxidant activities with DPPH (%) age inhibition of 84%, whereas ascorbic acid, taken as standard, shows 94% activity as shown in Figure 1A. This high antioxidant activity of *P. nigrum* seed extract may be owing to the presence of phenolic and flavonoid components in this plant.

Tyrosinase is an important enzyme which plays an important part in melanin synthesis in humans and is also found in animals. Therefore, it is considered a key enzyme in melanogenic studies in humans. As shown in Figure 1B, *P. *nigrum** seed extract has very good tyrosinase inhibition activity, as shown in Figure 1B in comparison with kojic acid. The anti-tyrosinase activity of *P. nigrum* seeds can be recognized by the existence of phenols, flavonoids, and antioxidant constituents reported in them [12].

In recent years, GC with MS detection (GC-MS) has been studied as a very important technical analysis platform for secondary metabolite profiles such as phytochemicals [16]. The present study has been designed to determine the phytochemical constituents of seed extract of *P. nigrum* by using GC-MS analysis. The results of GC-MS lead to the identification of plenty of compounds based on mass spectrum fragmentation patterns in the seed extract of *P. nigrum* as shown in Figure 2. Based on the results of GC-MS analysis as shown in Table 1, it is evident that there are mainly eighteen compounds with different retention times in the seed extract of *P. nigrum*. Beta-pinene 1, D-limonene 2, terpineol 3, 3-cyclohexene-1-methanol 4, caryophyllene 5, delta-cadinene 6, ((1S,2S,4R)-(-)-alpha, alpha-dimethyl-1-vinyl-o-menth-8-ene-4-methanol)/Elemol 7, copaene 8 and myristic acid 9, from our results, it was observed that caryophyllene, delta-cadinene, copaene, n-hexadecanoic acid and piperine were the major components in the seed extract of *P. nigrum*. Most of the biological activity data in Table 1 has been provided by Dr. Duke’s phytochemical and ethnobotanical databank. The phytochemicals in the seed extract of *P. nigrum* have been established to show fascinating biological activities such as antioxidant, anti-inflammatory, anticancer, analgesic, anesthetic, antiedemic, antimicrobial, antidermatitic, and antiacne properties. Antioxidant activity constitutes an important part of the biological activities of obtained phyto-components, and this situation is in accordance with our results of antioxidant activity of 84%. On the other hand, the anti-inflammatory effect is highly observed in 3-cyclohexene-1-methanol, caryophyllene, hexadecanoic acid methyl ester, piperine, and γ-tocopherol from the compounds determined in this study. Studies show that there is a great need for the use of chemical protective mediators with antioxidant, anti-inflammatory, and phenolic qualities to protect the skin against ultraviolet radiation [17,18]. The presence of antioxidant, anti-inflammatory, and phenolic phytochemicals detected in the study may play a vital part in safety against ultra-violet radiation. Caryophyllene, which is the major phyto-component in the GC-MS chromatogram, is the structure of a bicyclic sesquiterpenoid and has analgesic, antitumor, anti-inflammatory, antibacterial, anti-dermatitic, anti-acne biological activities based on Dr. Duke’s phytochemical and ethnobotanical databank [19,20]. It is noteworthy that isopropyl myristate (IPM), which is used in different applications as an emollient, thickening agent, or lubricant in the cosmetics and pharmaceutical industries, was also found according to GC-MS results [5]. IPM’s excellent spreading properties combined with fast and easy absorption by the skin make it a substitute for natural oils in the cosmetics industry [14]. The presence of γ-tocopherol, one of the different forms of vitamin E, was determined as a result of GC-MS analysis. Tocopherols have a very important place in the cosmetic industry as antioxidants and skin protectors to minimize skin harm caused by ultraviolet radiation and are also indicative of ultraviolet defensive qualities in the ultraviolet region [21,22]. All in all, many of the phyto-components detected by GC-MS in the seed extract of *P. nigrum* have valuable biological activities and are worth further investigation in product development for ultra-violet radiation-induced skin protection.

*In vitro* evaluation of formulations was carried out with emphasis on two main features that were analyzed, i.e., phase separation and liquefaction. These parameters (phase separation and liquefaction) were tested by retaining base and plant extract-containing formulations at various temperatures and RH levels (8 ± 1 °C, 25 ± 1 °C, 40 ± 1 °C, and 40 ± 1 °C + 75 ± 1% RH) for three months. The stability in formulations containing natural extract was due to the presence of natural antioxidants i.e., *P. nigrum* seeds, which prevent the formulation from oxidative deprivation and microbial development, which is the basis of the formulation color change. The other ingredients, such as emulsifying agents, also help to prevent color change [23]. The liquefaction is caused by the variation in temperature, which is the reason for the modification in the viscosity of the emulsion. The phase separation may be because of the difference in density among the phases of two immiscible systems [16]. Table 2 shows the results of the present study. Concerning liquefaction and phase separation, the control and extract-containing formulation were consistent and stable at various temperatures and RH conditions. There was no microbial growth at any stage for 90 days stability study. Moreover, in a recent study, the formulations were found to be stable owing to the high viscosity imparted by the gel phase of these gellified emulsion systems that entrap the globules of emulsion by making a three-dimensional network.

pH is a very important parameter in skin cosmetic products. The skin’s pH varies from 4.0 to 7.0. The ideal pH on average is 5.0 to 6.0 for cosmetic purposes [24]. As shown in Figure 3, there was a small decline in pH of both the placebo and test formulation during the three-month period of the study. This decrease in pH might be due to aldehydes, organic acids, and other metabolites which are produced during the decomposition of liquid paraffin [25].

The formulation stability is linked with the globule form and magnitude of the internal phase of the emulgel [26]. The miniscule evaluation of globule size and shape of the internal phase of emulgel is an important tool that is associated with formulation stability. Microscopic evaluation displayed that the emulsion globules developed in the gel bases of all emulgel formulations were of spherical shape, which is an indicator of a suitable and accurate method to prepare all the gellified emulsions as shown in Figure 4. Throughout the whole investigational period, the spherical shape of the globules remained intact, and no change in morphology was observed even after 90 days.

The electrical conductivity is used to measure the formulation stability. Conductivity is basically the parameter for the measurement of free ions [27]. In the current study, the conductivity reading of base and extract-containing formulations kept at various testing storage conditions (8 ± 1 °C, 25 ± 1 °C, 40 ± 1 °C, and 40 ± 1 °C + 75 ± 1% RH) was observed at a definite time interval for an investigational period of 90 days as shown in Figure 5. A gradual and slight increasing trend in electrical conductivity values was seen for emulgel formulations and placebo kept at various storage conditions. In this work, no sudden or sharp increase in electrical conductivity values was observed, so it might be assumed that all the studied emulgel formulations were stable within the applied conditions.

In the context of topically applied emulgels (semisolid dosage forms), the parameter of shear rate is of paramount importance [28]. In a recent study, a slight increase in the shear rate values of both test and placebo formulations was observed gradually with time and storage temperature in the rheograms between shear rate and shear stress as depicted in Figure 6 and Figure 7. The particles in a gelling system arrange themselves in the flow direction of the system when shear stress is applied. This arrangement of particles may cause a decrease in internal resistance to flow, so the final viscosity is also decreased. These different rheological parameters are helpful in analyzing emulgel stability by relating the changes that happen with increasing shear stress and other changes in temperature.

The centrifugation test is the main indicator for the assessment of shelf life of formulation [29]. It is based on the principle of separation of different components of the emulsion depending on their relative densities. This test was conducted at the time of formulation preparation of both the placebo and extract-containing formulation followed by 24 h and 7, 14, 28, 60, and 90 days. Both formulations remained stable after centrifugation during 90 days of stability studies.

Sun protection capacity is a measurable way to assess product efficacy against an inclusive series of ultra-violet radiation, which marks emulgel to be used as a sunblock to avoid sunburn and other skin harm. Phytochemicals such as tannins, flavonoids, and phenols are the main ingredients that protect the skin from ultraviolet radiation [30]. Thus, extracts containing these phyto-ingredients are ideal candidates for protecting skin from unwanted or damaging rays and may show potent natural sunblock in the ornamental formulation and therefore can possibly avoid skin damage. Thus, the sun protection assessment of emulgel formulation could be linked with the existence of these phenolic and flavonoid components in the plant abstract. In the current study, the formulations containing *P. nigrum* seed extract showed SPF values of 7.51. Hence, the skin can be protected against the deleterious effects of ultraviolet rays by applying these formulations as they may prove to possess sunscreen agents of natural origin.

Zeta potential is the charge that grows at the border between a dense exterior and its fluid medium. This potential, which is measured in millivolts, may arise by any of several mechanisms. The zeta potential gives an indication of the charge present on the particles’ surface [31]. The result of the zeta potential of the *P. nigrum* containing formulation was −45 mV. This negative sign shows the negative charge on globules. Table 3 and Figure 8 show the relationship between zeta potential and emulsion stability. Under investigation by zeta potential, the *P. nigrum* extract formulation falls into a fairly good stability category. These categories are classified in Table 3. Hence, it is clearly evident that the *P. nigrum*-containing formulation is stable.

The skin tolerance test is essential for topical formulations, and topical formulations should have high levels of skin tolerability for improved compliance during usage [32,33]. Determining the skin tolerance capabilities of the designed *P. nigrum* emulgel and plain gels is crucial. The rat that was given the irritating formaldehyde solution, however, had obvious skin disturbances, including severe erythema and maybe inflammatory cellular infiltrations. The biocompatibility of the excipients utilized in making the *P. nigrum* emulgel, particularly acetyl alcohol, liquid paraffin oil, Tween 20, span 80, methyl parabin, and propylene glycol, which are often regarded as safe excipients, may be the cause of the emulgels’ superior skin tolerability. According to the study, rat skin accepted the *P. nigrum* emulgel without developing a weeping surface, making topically applying it safe. The outcomes of this research also provide evidence in favor of the claim that this emulgel is secure and efficient for delivering *P. nigrum* to the skin.

A strong spreadability rating is one of the key requirements for an emulgel. Spreadability is a crucial component of treatment and is measured as an indication of application simplicity [34]. The formulation’s spreadability has a significant impact on how well the medicine is administered at the proper dosage. The distribution area of both control and formulation was expanded by applying load (weight). This might be due to the good gelling property of the gelling agent, carbacol 940, used in the formulations. Furthermore, the outstanding spreadability figures show that carbacol 940 is extremely compatible with the oil and other excipients employed in the formulations, and the gels are spreadable with little shear. The outcome supports the theory that the gels will be appropriate for skin application and may promote enhanced topical anti-aging action of *P. nigrum* since they are simple to apply to the skin.

The drug contents determination shows that the gel system built on carbacol 940 offered adequate space for the drug-loaded emulgel carrier system to be solubilized and trapped without any leakage, resulting in an outstanding overall drug content percentage. As a result of a favorable molecular interaction between *P. nigrum* and liquid paraffin oil, the results also indicated that the high solubility of *P. nigrum* in the liquid lipid tended to promote drug entrapment and, thus, affected drug content efficiency.

A cumulative drug release versus time graph of PNEF and PNCF was plotted, and it revealed that the *in vitro* release pattern is a crucial parameter for determining drug diffusion across the membrane, which is influenced by a number of factors including interactions between formulation and drug, physicochemical properties of ingredients and internal structure [35]. The slow release of the drug (*P. nigrum* bioactive components) from PNEF may be attributed to the high viscosity of PNEF as compared to PNCF, owing to the Carbopol gel. Earlier research studies have confirmed that *in vitro* drug release decreases with the increase in the viscosity of formulations [36]. Another explanation for delayed release from PNEF might be gel remodeling, which lengthens the active moiety’s diffusion pathway by decreasing the diffusion area. On application of paired sample *t*-test, results show a significant difference in drug release from PNEF and PNCF.

In silico studies have been effectively executed for the depiction of the forecast for ligand-target contact for an improved understanding of the molecular origin of the natural action of the original product [37]. This study provides an additional prediction as a probable tool to elucidate the binding affinity of a component in contrast to an enzyme. Four components from the GC-MS summary of ethanol *P. nigrum* extract (γ-tocopherol, campesterol, phytol, and myristic acid) were docked in comparison to tyrosinase enzyme to provide further insight into the plant’s inhibition capacity and to compare the experimental enzyme inhibition result. In silico molecular docking results show the binding between tyrosinase enzyme and ligands as shown in Table 5. γ-tocopherol, campesterol, phytol, and myristic acid as identified from GC-MS investigation, endorse the role of *P. nigrum* in tyrosinase inhibition assays as displayed in Figure 10 and Figure 11.

## 4. Materials and Methods

### 4.1. Chemicals and Equipment

Paraffin oil (Merck KGOA, Darmstadt, Germany), acetyl alcohol (St. Louis, MO, USA), span 80 (Sigma-Aldrich, Darmstadt, Germany), 2,2-diphenyl-1-picrylhydrazyl (DPPH) (Sigma-Aldrich, Darmstadt, Germany), propyl paraben (Sigma-Aldrich, Darmstadt, Germany), Tween 80 (Sigma-Aldrich, Darmstadt, Germany), propylene glycol (Sigma-Aldrich, Darmstadt, Germany), methyl paraben (Sigma-Aldrich, Darmstadt, Germany), ethyl alcohol (Ethanol) (Sigma-Aldrich, Darmstadt, Germany), carbopol-940 (Sigma-Aldrich, St. Louis, MO, USA), triethanol amine (TEA) (Sigma-Aldrich, Darmstadt, Germany) were used in this research work. Purified water was manufactured at the laboratory of cosmeceuticals, Department of Pharmacy, IUB, Pakistan. Rotary evaporator (Heidolph, Co., Ltd., Schwabach, Germany, Bioevopeak Inc., Seattle, WA, USA), UV spectrophotometer (Boschaplatz 3 Presseck, 95355 Germany), microplate reader Synergy HT (BioTek Instrument, Stevens Creek Blvd, Santa Clara, CA, USA), refrigerator (Dawlance, Karachi, Pakistan), hot incubator (Sanyo MIR-162, Moriguchi City, Japan), digital humidity meter (TES Electronic Corp., Taipei, Taiwan), pH meter (WTW pH-197i, Weilheim, Germany), optical microscope (Eclipse E200, Nikon, Melville, NY, USA), and conductivity meter (WTW COND-197i, Weilheim, Germany) were used.

### 4.2. Plant Material

*Piper nigrum* seeds were purchased from the native market of Multan, Punjab, Pakistan, and recognized by the Department of Botany, The Islamia University Bahawalpur, Pakistan, and a specimen was submitted with reference number 151.

### 4.3. Preparation of Extract

*Piper nigrum* seeds were washed with warm water to remove debris and other unwanted material, followed by shade drying. These dried seeds were ground into a powder with the help of a grinder and stored in a firmly closed bowl at 25 ± 1 °C to avoid any adulteration. The powder (150 g) was soaked in 80% ethanol for seven days. The initial filtration was performed with a muslin fabric followed by Whatman filter paper No.1. Finally, the mixture was concentrated by one-third of its original amount with a rotary evaporator at 44 ± 1 °C and the concentrated extract was stored in a refrigerator for further analysis [25].

### 4.4. Phytochemical Analysis

#### 4.4.1. Total Phenolic Content (TPC)

The free radical scavenging activity is related to the amount of phenolic acids. These phenolic compounds are the cause of antioxidant activity. In the present study, the TPC of the extract was measured by means of Folin–Ciocalteu reagent (FCR) according to Kim’s method with slight modification. Briefly, 1.0 mL of FCR was mixed into a combination comprising 1 mg of sample in 9 mL of water and blended completely for 5 min. Afterwards, 10 mL of Na_2_CO_3_ was mixed, and an exact quantity of 25 mL was prepared with distilled water. The combination was incubated at room temperature for 90 min, and absorbance was measured at 750 nm. TPC was expressed as milligrams of gallic acid equivalents per gram of extract (GAE/g of dry extract) [38].

#### 4.4.2. Total Flavonoids Contents (TFC)

Park’s method with minor modifications was used to estimate the TFC in the extract. The volume of 0.1 mL of 0.3 mol AlCl_3_ was mixed into a mixture containing 0.3 mL of plant extract and 0.5 mol/L NaNO_2_. Afterwards, 3.4 mL of 30% methanol was mixed with the above blend, and absorbance was recorded at 506 nm. TFC was stated as milligrams of quercetin equivalents per gram of extract (QE/g of dry extract) [38].

### 4.5. Antioxidant Activity by DPPH Method

The antioxidant activity of *P. nigrum* seed extract was measured with the DPPH method [10]. Briefly, 5 µL of the sample was mixed with 95 µL of 2, 2-diphenyl-1-picrylhydrazyl (DPPH) solution. The solution was mixed and incubated for 30 min at 37 ± 1 °C in a dark area. After the incubation, the absorbance was taken at 517 nm by taking ascorbic acid as a standard using a 96-well plate reader [28]. Equation (1) was used to measure the percentage scavenging effect.
Scavenging effect (%) = 100 − [(A° − A1/A°) × 100](1)
where A° is the absorbance of the control reaction and A1 is the absorbance in the presence of a sample of *P. nigrum* seed extract.

### 4.6. Tyrosinase Enzyme Inhibition Activity

The tyrosinase inhibition activity of the sample was evaluated by taking kojic acid as a positive control. The 60 units of the enzyme were mixed with 150 µL of buffer (50 mM of pH 6.8) and 40 µL of plant specimen compound in each well was incubated at 30 °C for 15 min. After incubation, pre-read was noted at 480 nm. Afterwards, 40 µL of 1 mM of the substrate (L-DOPA) per well was mixed and re-incubated at the same state for 30 min. After re-incubation, absorbance was taken again at the same wavelength [39].

### 4.7. Gas Chromatography-Mass Spectroscopy

GC-MS equipment (Agilent, 6890 series and Hewlett Packard, 5973 mass selective detector) was used. Separations were accomplished using an HP-5MS column (30 mm in length, 250 µm in diameter, and 0.25 µm in film thickness). Spectroscopic detection by GC–MS involved an electron ionization system that utilized high-energy electrons (70 eV). Injector temperature set to 220 °C; transfer line temperature set to 240 °C; oven temperature programmed to range from 60 to 246 °C, at a rate of 30 °C/min. Pure helium at a flow rate of 1 mL/min was used as a carrier gas. At 250 °C, 1.0 L of prepared extracts was injected in a split-less mode. The initial temperature was set at 50–150 °C with an increasing rate of 3 °C/min and a reading time of about 10 min. Finally, the temperature was raised to 300 °C at a rate of 10 °C/min. The detection was carried out in full scan mode between 35 and 600 m/z, and the bioactive compounds were identified using the National Institute of Standards and Technology (NIST) 2011 MS Library database [40].

### 4.8. Preparation of Test and Placebo Formulation

The gel base was manufactured by spreading carbapol 940 at 1000 rpm into a glass container containing double distilled water followed by the drop-wise addition of triethanolamine (TEA) to adjust the pH of the gel. The gel base was left for 24 h to construct the gel system. The emulsion was prepared by using different constituents as mentioned in Table 5. Before homogenization, the aqueous phase and oil phase were heated up to 75 ± 1 °C. After heating the aqueous phase and oil phase, the oil phase was mixed drop-wise into the aqueous phase with a stirring rate of 2000 rpm for 15 min, followed by slowing down to 1500 rpm for 10 min. To achieve complete homogenization, the speed was gradually reduced to 1000 rpm for 10 min and 500 rpm for 5 min. The prepared emulsion was refrigerated at room temperature. The emulsion was added to the reformulated gel phase to obtain the emulgel. The same method was executed for the control formulation [41].

### 4.9. Emulgel In Vitro Characterization

The control and *P. nigrum* containing emulgel were focused on to assess the stability studies for a 90 day (three months) period at accelerated conditions of temperature and humidity (8 ± 1 °C, 25 ± 1 °C, 40 ± 1 °C, and 40 ± 1 °C + 75 ± 1% RH) [42].

#### 4.9.1. Organoleptic Evaluation

Organoleptic parameters such as color, liquefaction, phase separation, and microbial growth were tested for a time period of three months by keeping both the base and *P. nigrum* formulations at various temperature conditions (8 ± 1 °C, 25 ± 1 °C, 40 ± 1 °C, and 40 ± 1 °C + 75± 1% RH).

#### 4.9.2. pH

pH change was measured for the period of three months by means of a pH meter (WTW pH-197i, Germany) by keeping both base and *P. nigrum* formulation at various temperature conditions (8 ± 1 °C, 25 ± 1 °C, 40 ± 1 °C, and 40 ± 1 °C + 75 ± 1% RH).

#### 4.9.3. Microscopy

Particle morphology and trend in size variation (increase or decrease) were measured for the period of three months by means of an optical microscope (Eclipse E200, Nikon, Japan) by keeping both base and *P. nigrum* formulation at various temperature conditions (8 ± 1 °C, 25 ± 1 °C, 40 ± 1 °C, and 40 ± 1 °C + 75 ± 1% RH).

#### 4.9.4. Electrical Conductivity

The changes in electrical conductivity were measured for the period of three months by using an electrical conductivity meter (WTW COND-197i, Germany by keeping both base and *P. nigrum* formulation at various conditions of temperature and RH (8 ± 1 °C, 25 ± 1 °C, 40 ± 1 °C, and 40 ± 1 °C + 75 ± 1% RH).

#### 4.9.5. Rheological Evaluation

Brookfield DVIII Ultra Rheometer (Brookfield engineering labs. Inc., Middleboro, MA, USA) was utilized for the assessment of viscosity, thixotropic, and flow properties of the emulgel. The apparatus was fitted with a CP41 spindle and an automatic temperature control system to get rheological data at 25 ± 1 °C. Shear stress, shear rate, viscosity, and thixotropic parameters of the emulgel formulations were evaluated according to the preset program of the rheometer. Spindle speed was increased gradually from 10–50 rpm with an increment of 10 rpm and again decreased to 50–10 rpm with a decrement of 10 rpm. The obtained rheograms comprising shear rate and shear stress were analyzed using Rheocalc software (V. 2.6) (Middleboro, MA 02346 USA) according to the Power law as the following Equation (2),
τ = K γ n(2)
where τ is shear stress, γ is shear rate, K is consistency index, and n is the flow index. The rheological evaluation was performed for each fresh emulgel formulation and followed at regular intervals throughout the study period stored at (8 ± 1 °C, 25 ± 1 °C, 40 ± 1 °C, and 40 ± 1 °C + 75 ± 1% RH).

### 4.10. Centrifugation

By keeping both base and *P. nigrum* formulation at various parameters (8 ± 1 °C, 25 ± 1 °C, 40 ± 1 °C, and 40 ± 1 °C + 75 ± 1% RH) the phase separation was assessed during three months study by using centrifugation machine [43].

### 4.11. In Vitro Sun Protection Factor (SPF)

*In vitro* SPF of the plant extract-containing formulation was measured by means of a spectrophotometer as reported previously. About 1 g of the test was dissolved in 100 mL of ethanol in a volumetric flask and sonicated for 5 min. Following sonication, it was passed through a cotton plug and the first 10 mL of this solution was rejected. Next 5 mL of this was again made to 50 mL with ethanol and 5 mL of which was made to 25 mL with ethanol. By taking ethanol as blank, the absorption of the sample was measured in the range of 290–320 nm every 5 nm. Mansur’s equation was used to calculate the SPF [43].

### 4.12. Particle Size and Zeta Potential

The particle size and zeta potential of optimized *P. nigrum* extract-containing formulation were assessed by dynamic light scattering using a Malvern zeta sizer at a temperature of 25 °C. Prior to measurements, samples were diluted with deionized water. Zeta potential values were presented as the mean ± standard error of the mean (SEM) from triplicate experiments [44].

### 4.13. Skin Irritation (Tolerance) Test

With a few minor modifications to the previously described procedure [45], the skin tolerance potential of *P. nigrum* emulgel was assessed. Four 180–200 g rats were used in this study, and the rat dorsal skin was hair-free shaved 24 h before the experiment. The animal’s dorsal side (1–2) was treated with *P. nigrum* emulgel (1 g), which was evenly placed across a 0.4 cm^2^ area and then taped up. A 1% formaldehyde solution was applied to the dorsal skin of the third rat as a positive control, and a placebo gel (plain gel) was applied to the fourth rat as a negative control. For three days, the gel was administered once daily, and the skin was examined for any obvious changes such as redness, oedema, and skin rash.

### 4.14. Spreadability

By measuring the spreading diameter of 0.5 g of emulgel that was placed within a circle that was 1 cm in diameter and pre-marked on a glass plate over which a second glass plate (75 gm) was placed, the spreadability of the emulgel formulations was ascertained 48 h after production. The upper glass plate was left with a weight of 425 g for 5 min, after which no further spreading was anticipated. It was observed that the gels’ spreading caused the diameter to grow [45]. The spreadability (g. cm. min^−1^) was calculated using Equation (3):S = m × l/t(3)
where: S is spreadability, m is the weight of the upper plate and rested on it (g), l is the diameter of the spreading emulgel (cm), and t is the time taken (min).

### 4.15. Drug Content Determination

One gram of emulgel was added in a suitable solvent and combined to get a clear solution and then filtered. Then, a UV spectrophotometer was utilized to ascertain its absorbance. The same solvent is used to prepare a standard drug plot. The same standard plot was used to determine concentration and drug content by using the absorbance value [34].

### 4.16. In Vitro Release

Drug (bioactive components) release studies were performed to compare the release pattern from *P. nigrum* loaded emulgel and without *P. nigrum* loaded emulgel (control emulgel). For release studies, the Franz diffusion cell (EMFDC-06) was used. The Franz cells were equipped with a cellophane dialysis tubing membrane (MWCO 12–14000 Da) having a surface area of 1.76 cm^2^, which was soaked for 24 h in an acceptor medium at 32.0 ± 0.5 °C, prior to use in studies. The temperature of the water bath and donor compartment was maintained at 32–37 °C. About 0.5 g of each gel (control and *p. nigrum* loaded emulgel) was placed and spread evenly on the membrane via the donor compartment. The temperature of the donor compartment was maintained under thermostatic conditions by using a multi-magnetic stirrer. Two milliliters of sample was taken at predetermined time intervals, i.e., 0, 0.5, 1, 2, 3, 4, 5, 6, and 12 h, and the same volume was replaced with BPS (phosphate buffer solution) pH 5.5 for 12 h studies. A UV-spectrophotometer was used to analyze the sample at 280 nm and the amount of cumulative drug release versus the time graph of the *P. nigrum* loaded emulgel and without *P. nigrum* loaded emulgel (control emulgel) was plotted [46].

### 4.17. In-Silico Molecular Docking Studies

Molecular docking was performed to assess theoretical binding attraction and interaction between ligand and tyrosinase target enzyme. The tyrosinase enzyme was downloaded from the Protein Data Bank [47]. The ligands were downloaded from the PubChem database. For this docking computation PyRx and for visualization of docking results Discovery Studio 2021 client was used [48].

### 4.18. Statistical Analysis

For statistical analysis, SPSS version 23.0 (Chicago, IL, USA) was used to evaluate the *in vitro* parameters.

## 5. Conclusions

We report a stable emulgel system containing *P. nigrum* seed extract (4%) with excellent antioxidant activity and sun protection abilities due to phenolic content in the extract that can be pivotal in protecting the skin from dangerous ultraviolet sun rays. Furthermore, the control or base and *P. nigrum* seed extract-containing preparations remained consistently stable over three months at various temperatures and RH conditions (8 ± 1 °C, 25 ± 1 °C, 40 ± 1 °C, and 40 ± 1 °C + 75 ± 1% RH). The skin tolerance test showed the safety of the skin. Thus, the prepared emulgel formulation can be used as a natural substitute for skin safeguard in comparison to preparations with synthetic components. Moreover, non-invasive in vivo investigations are required to validate the potency of *P. nigrum* seed extract emulgel for its use on human skin to evade the skin grievances resulting from electromagnetic emission.

## Figures and Tables

**Figure 1 molecules-27-05990-f001:**
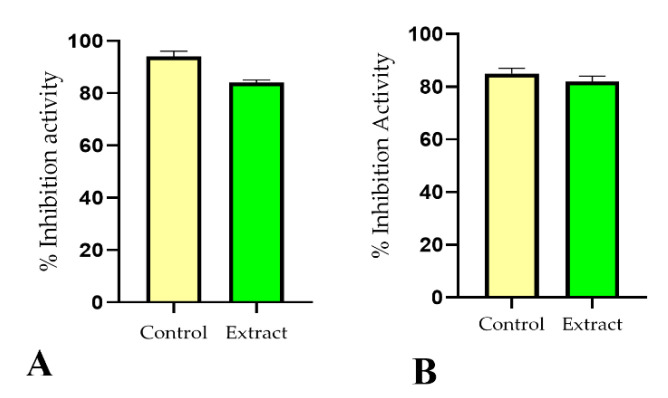
Antioxidant activity (**A**) and anti-tyrosinase activity (**B**) of *P. nigrum* extract (mean ± SD, *n* = 3).

**Figure 2 molecules-27-05990-f002:**
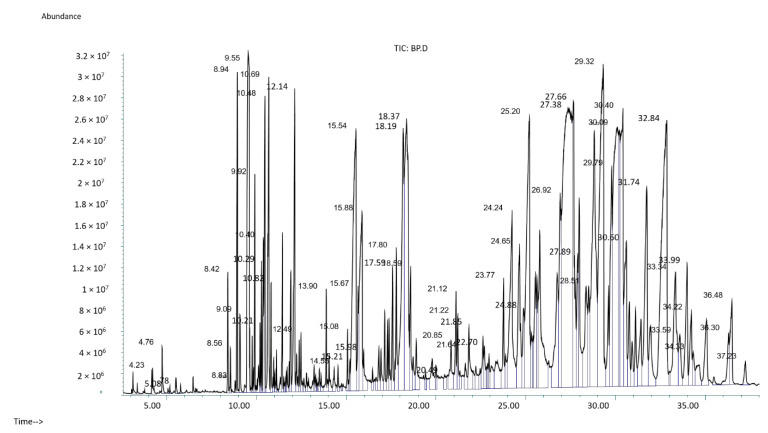
GC-MS spectra of ethanolic extract of *P. nigrum*.

**Figure 3 molecules-27-05990-f003:**
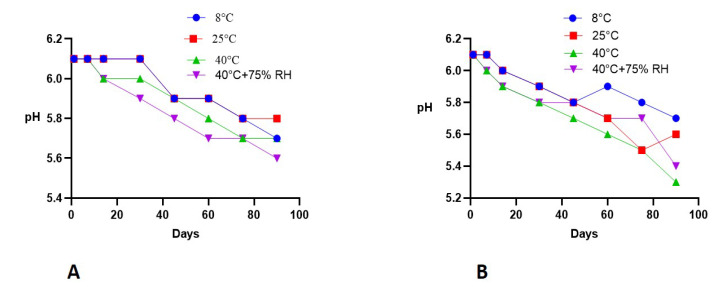
pH changes of (**A**) (placebo) and (**B**) (formulation) with time kept at different storage conditions (mean ± SD, *n* = 3).

**Figure 4 molecules-27-05990-f004:**
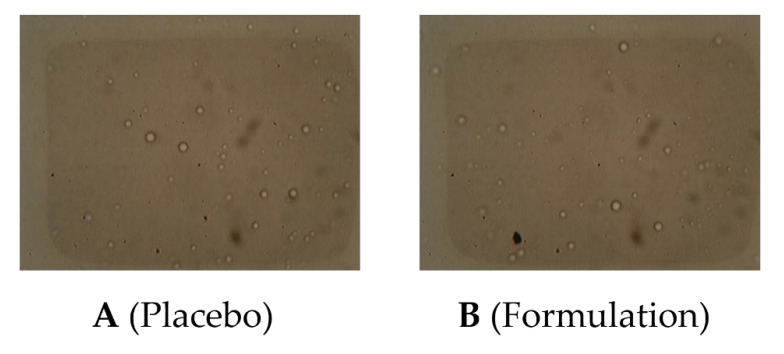
Globule shape of freshly prepared base (**A**) and test (**B**).

**Figure 5 molecules-27-05990-f005:**
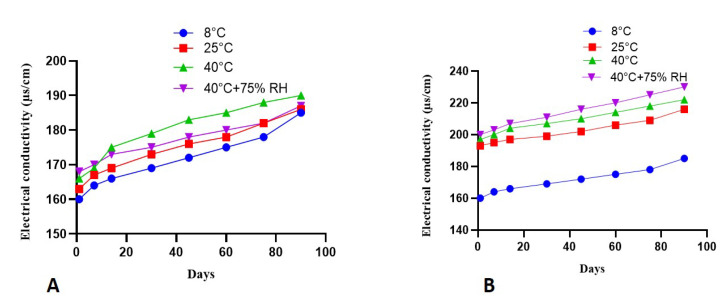
Change in electrical conductivity of (**A**) (placebo) and (**B**) (formulation) with time kept at different storage conditions (mean ± SD, *n* = 3).

**Figure 6 molecules-27-05990-f006:**
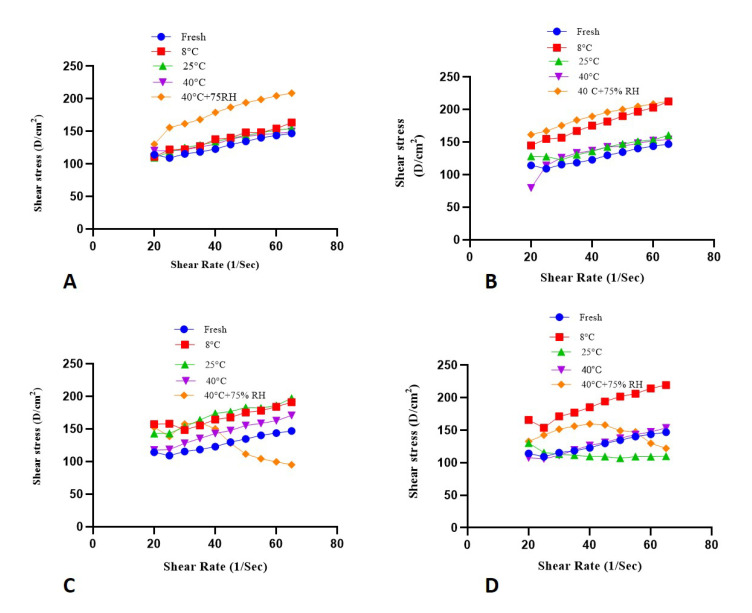
Rheograms of placebo kept at (**A**) (fresh), (**B**) (after 30 days), (**C**) (after 60 days), (**D**) (after 90 days).

**Figure 7 molecules-27-05990-f007:**
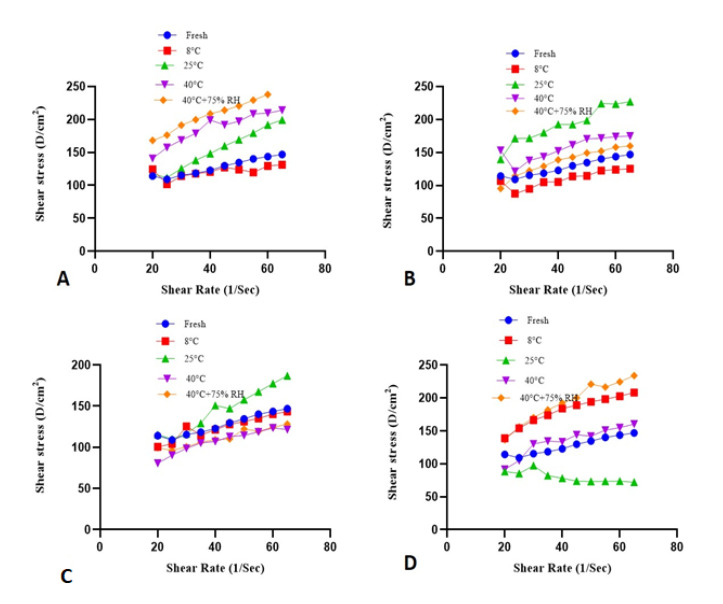
Rheograms of formulation kept at (**A**) (fresh), (**B**) (after 30 days), (**C**) (after 60 days), (**D**) (after 90 days).

**Figure 8 molecules-27-05990-f008:**
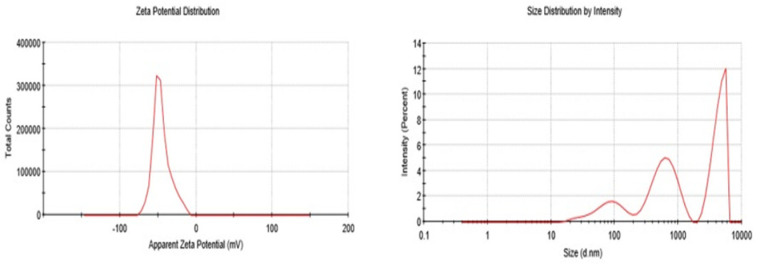
Graphical representation of zeta potential and globule size.

**Figure 9 molecules-27-05990-f009:**
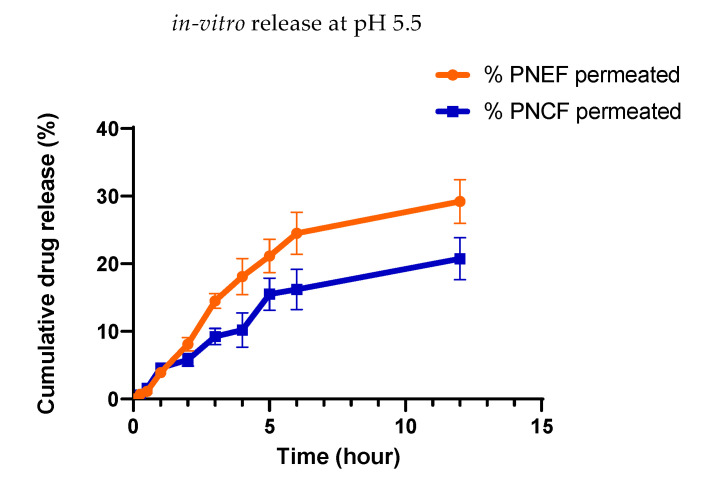
Comparison of % cumulative drug release from PNEF and PNCF after 12 h. Results represent mean ± standard deviation (indicated by vertical error bars), *n* = 3. There was a prominent difference in release rate between the PNEF and PNCF, as PNEF shows greater release in 12 h.

**Figure 10 molecules-27-05990-f010:**
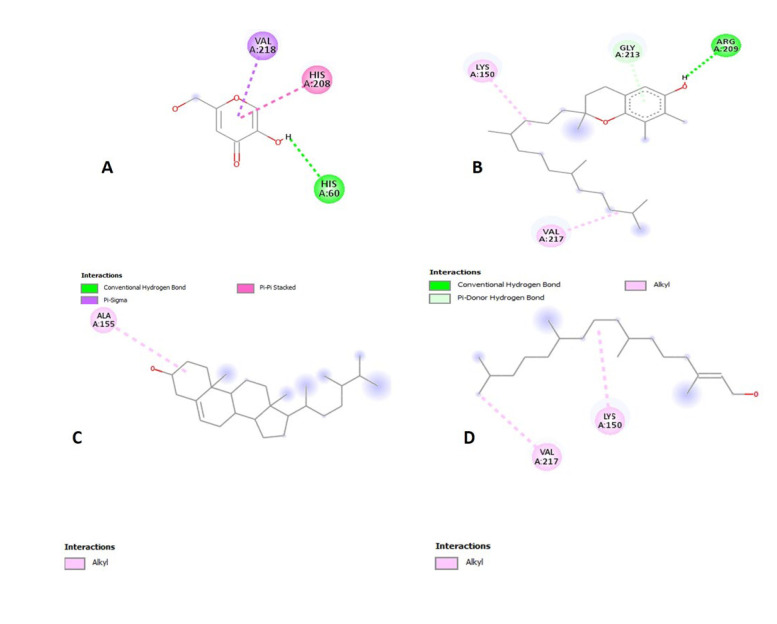
Interaction of tyrosinase ligands. (**A**) kojic acid, (**B**) γ-tocopherol, (**C**) campesterol, (**D**) phytol, and (**E**) myristic acid.

**Figure 11 molecules-27-05990-f011:**
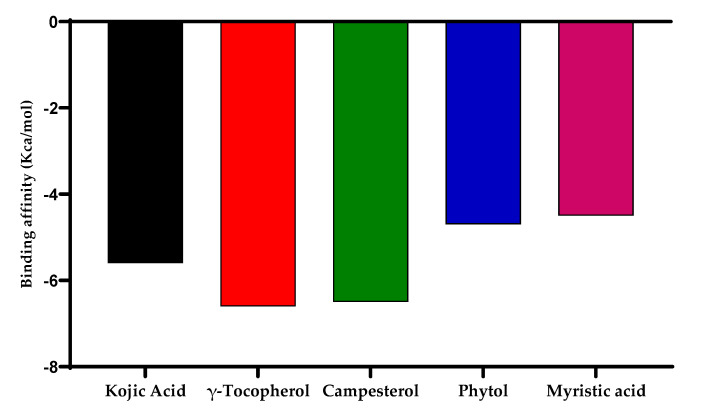
The binding affinity of kojic acid, γ-tocopherol, campesterol, phytol, and myristic acid.

**Table 1 molecules-27-05990-t001:** Gas chromatography-mass spectroscopy (GC-MS) analysis of phyto-compounds in the ethanolic *P. nigrum* seed extract.

Sr.	R_t_ (min)	Peak Area (%)	Name of Compound	Molecular Formula	MW	Nature of Compound	SI	Biological Activity
1	4.76	0.11	D-limonene	C_10_H_16_	136.23	Natural cyclic monoterpene	97	* Anticancer, Chemopreventive
2	5.54	0.05	Terpineol	C_10_H_18_O	154.25	Terpineol	94	* Insectifuge, Pesticide
3	6.63	0.02	3-cyclohexene-1 methanol	C_12_H_20_O_2_	196.29	Menthane monoterpenoids	92	* ACE-Inhibitor, Aldose-Reductase, Antibacterial, Antiinflammatory
4	9.55	2.87	Caryophyllene	C_15_H_24_	204.35	Bicyclic sesquiterpenoid	99	* Antitumor, Anti-inflammatory, Antibacterial, Antidermatitic, Antiacne
5	10.69	1.22	Delta-cadinene	C_15_H_24_	204.35	Bicyclic sesquiterpenoid	96	* Antibacterial, Anticariogenic,
6	10.97	0.08	(1S,2S,4R)-(−)-alpha,alpha-dimethyl-1-vinyl-o-menth-8-ene-4-methanol)/Elemol	C_15_H_26_O	222.37	Sesquiterpenoids, Tertiary alcohol	91	* Anti acetylcholinesterase, Antiulcer
7	12.14	1.22	Copaene	C_15_H_24_	204.35	Sesquiterpene	96	* Carminative
8	13.22	0.07	Myristic acid	C_14_H_28_O_2_	228.37	Long-chain fatty acids	99	* Antioxidant, Cosmetic, Cancer-Preventive, Lubricant
9	13.90	0.29	Isopropyl myristate	C_17_H_34_O_2_	270.50	Fatty acid ester	96	Emollient, Thickening agent, Lubricant [9]
10	15.08	0.14	Hexadecanoic acid methyl ester	C_17_H_34_O_2_	270.45	Fatty acid methyl esters	99	Anti-inflammatory [10], Flavor [11], Antioxidant [11], Antibacterial [12]
11	15.88	1.77	N-hexadecanoic acid	C_16_H_32_O_2_	256.42	Long-chain fatty acids	96	* Antialopecic, Antioxidant, Flavor,
12	17.39	0.26	8,11-octadecadienoic acid, methyl ester	C_19_H_34_O_2_	294.50	Ester	99	Not activity reported
13	17.59	0.46	Phytol	C_20_H_40_O	296.50	Acyclic diterpenoids	93	It produces anxiolytic and sedative effects [13], Antioxidant [14]
14	18.74	0.20	2-cis,6-trans-farnesol	C_15_H_26_O	222.36	Sesquiterpenoids	93	Not activity reported
15	26.76	1.37	Piperine	C_17_H_19_NO_3_	285.34	Alkaloids	99	* Analgesic, Antibacterial, Antiinflammatory
16	31.12	0.42	γ-tocopherol	C_28_H_48_O_2_	416.70	Methylated phenol	94	* Antioxidant, Anticancer, Antiinflammatory
17	34.33	0.24	Campesterol	C_28_H_48_O	400.70	Phytosterol	94	* Antioxidant, Hypocholesterolemic

R_t_ (min): Retention time, Peak area (%): It is the calculated percent area of the selected component compared to the total peak area in the chromatogram. SI (similarity index): For each component in the chromatogram, the spectra stored in the NIST library database are compared, SI: 100 when 100% matched, SI: 0 if completely different. * The biological activities of *P. nigrum* extract showing as taken from the Duke’s phytochemical and ethnobotanical database.

**Table 2 molecules-27-05990-t002:** Physical features of the fresh emulgel and after 90 days at different storage conditions (8 ± 1 °C, 25 ± 1 °C, 40 ± 1 °C, and 40 ± 1 °C + 75 ± 1% RH *).

Features	Base		Test Formulation	
	Fresh (0 h)	After 3 Months	Fresh (0 h)	After 3 Months
Color	White	White	Off white	Off white
Liquification	No	No	No	No
Microbial growth	No	No	No	No
Phase separation	No	No	No	No
Centrifugation	Stable	Stable	Stable	Stable

* RH = relative humidity and No = absence.

**Table 3 molecules-27-05990-t003:** Relationship between zeta potential and emulsion stability.

Stability Characteristics	Zeta Potential (mV)
Maximum agglomeration and precipitation	0 to +3
Range of strong agglomeration and precipitation	+5 to −5
Threshold of agglomeration	−10 to −15
Threshold of delicate depression	−16 to −30
Moderate stability	−31 to −40
Fairly good stability	−41 to −60
Very good stability	−61 to −80
Extremely good stability	−81 to −100

**Table 4 molecules-27-05990-t004:** The binding affinity of ligands and tyrosinase enzyme.

Compound Name	Binding Energy (kcal/mol)	Interacting Ligands at Binding Site of Enzyme	
		Bonding Type	Binding Amino Acid
Kojic acid	−5.4	Hydrogen bonding Pi-Pi stacked Pi-Sigma	His60 His208 Val218
γ-tocopherol	−6.6	Hydrogen bonding Alkyl	Arg209, Gly213 Lys150, Val217
Campesterol	−6.5	Alkyl	Ala155
Phytol	−4.7	Alkyl	Lys150, Val217
Myristic acid	−4.5	Hydrogen bonding Alkyl	Arg280, Glu289, Arg165, Val168, Ley169, Ile237, Val240, Ile243, Ile288

**Table 5 molecules-27-05990-t005:** Components of control and test formulation.

Components	Control (*w*/*w*)	Test (with Plant Extract) (*w*/*w*)	Role of Ingredients
Cetyl alcohol	1.50%,	1.50%,	Emulsifier
Liquid paraffin	9.00%	9.00%	Dispersing agent
Span 80	1.20%	1.20%	Surfactant
Propyl paraben	0.06%	0.06%	Preservative
Tween 80	1.92%	1.92%	Co-surfactant
Propylene glycol	10.00%	10.00%	Permeation enhancer
Methyl paraben	0.11%	0.11%	Preservative (prevent germ growth)
Double distilled water	76.21%	76.21%	Solvent
*Piper nigrum* extract	0.00%	4.00%	Active ingredient

## Data Availability

Not applicable.

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
