# Peer review of "Chemical Profiling, Formulation Development, In Vitro Evaluation and Molecular Docking of Piper nigrum Seeds Extract Loaded Emulgel for Anti-Aging"

_molecules, 2022, doi:10.3390/molecules27185990_

Round 1

Reviewer 1 Report

The authors describe the manuscript entitled " Chemical Profiling, Formulation Development, In-vitro Evaluation and Molecular Docking of Pipper nigrum Seeds Extract Loaded Emulgel for Anti-aging."

The idea of the work is good, but many Grammarly and typographical mistakes are noticed.

Firstly, the manuscript should be written in good language and clear of typos or punctuation mistakes.  

For example, author wrote anti-oxidant one time, and another wrote antioxidant, so please uniform your language.

Abstract

Check lines 31-32 and 40-42

Introduction

Check lines 46-85 and 91-95

Results and discussion

Merge results and discussion in one section

Check lines 104, 108, 110, 118-127, 129, 133-136, 139-140, rephrase 154-156, 217-237, 234-373, and caption figure 2

Complete check of all manuscript

In methodology, insilico studies which protein did you work in and why?

Author Response

Response to comments Manuscript ID: molecules-1839138 entitled

Article: “Chemical Profiling, Formulation Development, In-vitro Evaluation and Molecular Docking of Pipper nigrum Seeds Extract Loaded Emulgel for Anti-aging"

Response: Thank you for your kind observation. Also, thank you so much for your valuable insight on improving the manuscript. We greatly appreciate the reviewers’ comments and constructive suggestions. Those valuable comments are constructive to improve our manuscript and provide meaningful guidance for our future research. We hope this revised manuscript will meet the satisfaction of the reviewers. We would like to express our thanks to the reviewer for their comments and suggestions, which have improved our manuscript. We again appreciate your helpful suggestions. If you have any further suggestions for changes, please let us know.

As per your precious recommendation, we have made changes in our manuscript

Comments and Suggestions: The idea of the work is good, but many Grammarly and typographical mistakes are noticed. Firstly, the manuscript should be written in good language and clear of typos or punctuation mistakes. For example, author wrote anti-oxidant one time, and another wrote antioxidant, so please uniform your language.

Abstract

Check lines 31-32 and 40-42

Introduction

Check lines 46-85 and 91-95

Results and discussion

Merge results and discussion in one section

Check lines 104, 108, 110, 118-127, 129, 133-136, 139-140, rephrase 154-156, 217-237, 234-373, and caption figure 2

Complete check of all manuscript

In methodology, insilico studies which protein did you work in and why?

Response: The article was read thoroughly and improved by using some English grammar tools and also some other necessary changes are performed. The typographical mistakes were corrected. The manuscript was thoroughly checked and cleared punctuation mistakes.

The abstract lines 31-32 and 40-42 were checked and corrected.

The introduction lines 46-85 and 91-95 were checked and corrected.

Complete check of all manuscript was done.

For in silico docking studies, mushroom origin pdb 2y9x was used as protein. It was used after necessarily preparatory steps. The reason for the selection of this protein was that during in vitro enzyme inhibition assay, mushroom tyrosinase was used.

Reviewer 2 Report

The article describes the Piper nigrum extract entrapment in emulgel.

Although the methods and experiments are valid the manuscript has serious flaws. 

First, there is no exact scope of the work. If the work has been done in order to prepare a cosmetic formulation of Piper nigrum extract, there are definitely elements missing from the development. Stabilizers, preservatives, emollients etc are ingredients that are essential for a cosmetic formulation, that anyways cannot rely on just one extract as a bioactive ingredient. In the case that it is a topical pharmaceutical preparation things change, but this is not the case here, as there is no such experiment design. In this case introductory phrases as .. Topical drug delivery system has many advantages some  of which includes is to bypass first pass metabolism, escaping of risk associated with venous treatment, varied condition of absorption.. should be omitted.

Second, there is a confusion between topical cosmetic and pharmaceutical delivery. It is stated .. Emulgels also enhance the percutaneous penetration of active ingredient.. 

This is something not only non-desired in cosmetics, but it is against the definition of a cosmetic preparation.

Authors should decide for a concrete scope of the work.

Third, the works lacks innovation. It is basic implementation of already known techniques. Even in this case the paper could be acceptable if further biological investigation took place, through in vitro experiments.

The most prominent flaw of the manuscript is the bad English language, in terms of syntaxis and grammar, that needs a very thorough revision.

Author Response

Response to comments Manuscript ID: molecules-1839138 entitled "Chemical Profiling, Formulation Development, In-vitro Evaluation and Molecular Docking of Pipper nigrum Seeds Extract Loaded Emulgel for Anti-aging"

Comments: First, there is no exact scope of the work. If the work has been done in order to prepare a cosmetic formulation of Piper nigrum extract, there are definitely elements missing from the development. Stabilizers, preservatives, emollients etc are ingredients that are essential for a cosmetic formulation that anyways cannot rely on just one extract as a bioactive ingredient. In the case that it is a topical pharmaceutical preparation things change, but this is not the case here, as there is no such experiment design. In this case introductory phrases as .. Topical drug delivery system has many advantages some  of which includes is to bypass first pass metabolism, escaping of risk associated with venous treatment, varied condition of absorption.. should be omitted.

Response: Thank you for your kind observation and appreciating comments. Also, thank you so much for your valuable insight on improving the manuscript. We greatly appreciate the reviewers’ comments and constructive suggestions. Those valuable comments are constructive to improve our manuscript and provide meaningful guidance for our future research. We hope this revised manuscript will meet the satisfaction of the reviewers. We would like to express our thanks to the reviewer for their comments and suggestions, which have improved our manuscript. We again appreciate your helpful suggestions.

Pipper nigrum loaded emulgel was not prepared earlier and one respected reviewer appreciated this work. The role of ingredients in formulation was highlighted in table 1 which include Emulsifier,  Dispersing agent, Surfactant Preservative Co-surfactant, Permeation enhancer, Preservative (prevent germ growth), Solvent which stable the formulation as proved by various in-vitro test. In future for commercial usage the formulation can be further improved by addition of essences, preservative and whitening agent. The sentence in introduction phrase has been omitted.

Comments: Second, there is a confusion between topical cosmetic and pharmaceutical delivery. It is stated.. Emulgels also enhance the percutaneous penetration of active ingredient.. This is something not only non-desired in cosmetics, but it is against the definition of a cosmetic preparation. Authors should decide for a concrete scope of the work.

Response: Pipper nigrum loaded emulgel is a topical cosmetic delivery system, not a pharmaceutical delivery system. In case of topical cosmetic delivery system a less amount of active ingredient was penetrated into systemic circulation, which will show reduced side effects and was retained in an increased amount in the skin, while in the case of pharmaceutical delivery there is vice versa. To avoid confusion, the introduction phrase has been updated and the statement "Emulgels also enhance the percutaneous penetration of active ingredients." has been omitted from the introduction.

Comments: Third, the works lacks innovation. It is basic implementation of already known techniques. Even in this case the paper could be acceptable if further biological investigation took place, through in vitro experiments.

Response: Pipper nigrum loaded emulgel was not prepared earlier.  Four further in vitro experiments 1) Skin tolerance test 2) Spreadbility test 3) Drug Content Determination 4) In Vitro release were performed and described in article and in future its non-invasive in-vivo investigations will be performed.

Reviewer 3 Report

1. introduction lacks the information regarding the reported study of piper nigrum extract.

2. what is the rationale of preparing extract emulgel.

3. it lacks the standard marker compound data in the activity section.

4. why there is multiple peaks in size of formulations.

5. why only one formulation prepared. it must be optimized with different composition.

6. for which routes of administration it is prepared.

7. Its release must be performed.

Author Response

Response to comments Manuscript ID: molecules-1839138 entitled "Chemical Profiling, Formulation Development, In-vitro Evaluation and Molecular Docking of Pipper nigrum Seeds Extract Loaded Emulgel for Anti-aging"

Response: Thank you for your kind observation. Also, thank you so much for your valuable insight on improving the manuscript. We greatly appreciate the reviewers’ comments and constructive suggestions. Those valuable comments are constructive to improve our manuscript and provide meaningful guidance for our future research. We hope this revised manuscript will meet the satisfaction of the reviewers. We would like to express our thanks to the reviewer for their comments and suggestions, which have improved our manuscript. We again appreciate your helpful suggestions.

Major Issues

Comment 1: introduction lacks the information regarding the reported study of piper nigrum extract

Response: The introduction has been re-evaluated and included reported study piper nigrum extract.

Comment 2: what is the rationale of preparing extract emulgel.

Response: The rationale for emulgel includes that it has characteristics of both an emulsion and a gel and can be smoothly applied to the skin. Therefore, the active ingredient is easily incorporated into the emulgel and it shows good stability. As compared to other drug delivery systems, Emulgel is a good choice for topical cosmetic drugs.

Comment 3: it lacks the standard marker compound data in the activity section.

Response: The biological activities of Pipper nigrum extract showing as taken from the Duke's phytochemical and ethnobotanical database.

Comment 4: why there is multiple peaks in size of formulations.

Response: The GC-MS analysis was performed on a plant extract rather than a formulation. The multiple peaks may be of secondary metabolites present in the plant extract because this extract was not a pure plant drug.

Comment 5: why only one formulation prepared. It must be optimized with different composition.

Response: The formulation was optimized with different composition like changing in concentration of Pipper nigrum extract like 2% w/w, 3% w/w, 4% w/w but the formulation was stable and optimized at 4% concentration.

Comment 6: for which routes of administration it is prepared

Response: It is a topical drug delivery system. The most important route is gels which provide faster drug release compared with other topical drug delivery.

Comment 7: Its release must be performed.

Response: Its released was performed and included in article.

Round 2

Reviewer 2 Report

Dear authors

Although the English language was improved in the revised version, yet thorough revision is needed in order to reach proper scientific srandard.

Some points:

Line 70 

The components of emulgel contribute to the enhancement of drug bioavailability from these advanced delivery systems

Please notice that generally emulgels are not considered advanced delivery systems according to the literature. If you have access to a reference showing otherwise please insert.

Lines 618-637

It is not clear what is measured by UV spectroscopy in encapsulation and in vitro release assessment. Please add the thorough method.

Figure 10.

What is the standard deviation? Was the experiment done in triplicates? Statistical assessment?

Author Response

Response to Reviewer 2 Comments

Dear reviewer,

We greatly appreciated your careful reading of our manuscript and your valuable comments, which enormously allowed us to improve our manuscript quality. We have carefully considered the comments and have revised and rewritten the manuscript accordingly. Thank you very much for your time and the careful review. The revised manuscript was marked up with the changed text by highlighting them in yellow.

Comments and Suggestions for Authors

Dear authors

Although the English language was improved in the revised version, yet thorough revision is needed in order to reach proper scientific standard.

Response: Thank you very much for your positive feedback. We hope the following corrections and responses to your comments and suggestions satisfy you requests. The manuscript has been revised by a native English speaker. As well, the manuscript was thoroughly checked to reach the proper scientific standard.

Line 70

The components of emulgel contribute to the enhancement of drug bioavailability from these advanced delivery systems

Please notice that generally emulgels are not considered advanced delivery systems according to the literature. If you have access to a reference showing otherwise please insert

Response: The comment is much appreciated. Emulgel is not an advanced delivery system. The sentence "The components of emulgel contribute to the enhancement of drug bioavailability from these advanced delivery systems." has been omitted from the manuscript. Emulgel combines the characteristics of both emulsion and gel.

Lines 618-637

It is not clear what is measured by UV spectroscopy in encapsulation and in vitro release assessment. Please add the thorough method

Response: Response: Thanks for your comment. The in vitro release is a very important parameter, as it is helpful to measure drug diffusion across the membrane. In the present study, Piper nigrum loaded emulgel shows greater release as compared to control emulgel as shown in figure 9. As suggested, the method has been explained in more detail in the manuscript.

Figure 10. What is the standard deviation? Was the experiment done in triplicates? Statistical assessment?

Response: Thanks for your comment. Standard deviation is a measure of the amount of variation or dispersion of a set of values. Yes, the experiment was done in triplicates. The new graph has been incorporated into the manuscript after statistical assessment (Figure 9).

Finally, we would like to express our sincere thanks for the appreciated comments and suggestions that helped us enhance our manuscript’s quality.

Reviewer 3 Report

Accept

Author Response

The authors are thankful to the reviewer for appreciating our manuscript